# Multimodality Imaging in Ischemic Chronic Cardiomyopathy

**DOI:** 10.3390/jimaging8020035

**Published:** 2022-02-01

**Authors:** Giuseppe Muscogiuri, Marco Guglielmo, Alessandra Serra, Marco Gatti, Valentina Volpato, Uwe Joseph Schoepf, Luca Saba, Riccardo Cau, Riccardo Faletti, Liam J. McGill, Carlo Nicola De Cecco, Gianluca Pontone, Serena Dell’Aversana, Sandro Sironi

**Affiliations:** 1Department of Radiology, Istituto Auxologico Italiano IRCCS, San Luca Hospital, University Milano Bicocca, 20149 Milan, Italy; 2Department of Cardiology, Division of Heart and Lungs, Utrecht University, Utrecht University Medical Center, 3584 Utrecht, The Netherlands; m.guglielmo@umcutrecht.nl; 3Department of Radiology, Azienda Ospedaliero Universitaria (A.O.U.), di Cagliari-Polo di Monserrato, 09042 Cagliari, Italy; aserra@unica.it (A.S.); lucasabamd@gmail.com (L.S.); riccardocau00@gmail.com (R.C.); 4Radiology Unit, Department of Surgical Sciences, University of Turin, 10124 Turin, Italy; marcogatti17@gmail.com (M.G.); riccardo.faletti@unito.it (R.F.); 5Department of Cardiac, Neurological and Metabolic Sciences, Istituto Auxologico Italiano IRCCS, San Luca Hospital, University Milano Bicocca, 20149 Milan, Italy; valevolpato@hotmail.it; 6Department of Radiology and Radiological Science, MUSC Ashley River Tower, Medical University of South Carolina, 25 Courtenay Dr, Charleston, SC 29425, USA; schoepf@musc.edu (U.J.S.); mcgilll@musc.edu (L.J.M.); 7Division of Cardiothoracic Imaging, Nuclear Medicine and Molecular Imaging, Department of Radiology and Imaging Sciences, Emory University, Atlanta, GA 30322, USA; carlo.dececco@emory.edu; 8Centro Cardiologico Monzino IRCCS, 20138 Milan, Italy; gianluca.pontone@cardiologicomonzino.it; 9Department of Radiology, Ospedale S. Maria Delle Grazie—ASL Napoli 2 Nord, 80078 Pozzuoli, Italy; dellaversanaserena@gmail.com; 10School of Medicine and Post Graduate School of Diagnostic Radiology, University of Milano-Bicocca, 20126 Milan, Italy; sandro.sironi@unimib.it; 11Department of Radiology, ASST Papa Giovanni XXIII, 24127 Bergamo, Italy

**Keywords:** chronic ischemic cardiomyopathy, multimodality imaging, computed tomography angiography, echocardiography, cardiac magnetic resonance, nuclear medicine

## Abstract

Ischemic chronic cardiomyopathy (ICC) is still one of the most common cardiac diseases leading to the development of myocardial ischemia, infarction, or heart failure. The application of several imaging modalities can provide information regarding coronary anatomy, coronary artery disease, myocardial ischemia and tissue characterization. In particular, coronary computed tomography angiography (CCTA) can provide information regarding coronary plaque stenosis, its composition, and the possible evaluation of myocardial ischemia using fractional flow reserve CT or CT perfusion. Cardiac magnetic resonance (CMR) can be used to evaluate cardiac function as well as the presence of ischemia. In addition, CMR can be used to characterize the myocardial tissue of hibernated or infarcted myocardium. Echocardiography is the most widely used technique to achieve information regarding function and myocardial wall motion abnormalities during myocardial ischemia. Nuclear medicine can be used to evaluate perfusion in both qualitative and quantitative assessment. In this review we aim to provide an overview regarding the different noninvasive imaging techniques for the evaluation of ICC, providing information ranging from the anatomical assessment of coronary artery arteries to the assessment of ischemic myocardium and myocardial infarction. In particular this review is going to show the different noninvasive approaches based on the specific clinical history of patients with ICC.

## 1. Introduction

Ischemic chronic cardiomyopathy (ICC) still represents one of the most common cardiac diseases that can progress to acute myocardial ischemia or infarction, left ventricle (LV) dysfunction, or heart failure [1]. 

Currently, a multimodality approach in ICC is appropriate for the correct assessment and management of these patients considering that each technique can have an important role for planning a correct therapeutic strategy [2,3,4]. 

Following the ESC guidelines, the role of coronary computed tomography angiography (CCTA) is rapidly increasing given the high negative predictive value of this technique in patients with suspected coronary artery disease (CAD) [1]. Additionally, CCTA allows for the analysis of coronary plaque composition, severity of stenosis and segments involved. This information can be used to determine prognostic stratification [5,6]. 

However, given the low positive predictive value of CCTA [7], further CT imaging techniques such as fractional flow reserve CT (FFRct) and computed tomography perfusion (CTP) could represent an important tool for the depiction of hemodynamically significant CAD improving the diagnostic accuracy of CCTA alone [8]. 

Cardiac Magnetic Resonance (CMR) represents another important noninvasive tool for the assessment of patients with ICC [2]. A single examination can provide information regarding viability and inducible ischemia [2]; in particular the stress CMR is quickly moving to a quantitative assessment of myocardial ischemia overcoming the dichotomous assessment of stress CMR [2]. 

Considering the low cost and wide distribution of technique, echocardiography remains the first technique to be used in the patient with ICC. There are multiple advantages of echocardiography compared to other imaging techniques, including its safety, relatively low-cost, and portability which allows for bedside evaluation. 

According to the European guidelines, rest echocardiography is recommended in all patients in the initial diagnostic management of patients with suspected coronary artery disease (Class I level B) [1]. Echocardiography in this setting provides crucial information on cardiac anatomy, systolic and diastolic function, regional wall motion abnormalities, and associated valvular heart disease. Moreover, it gives precious information for the exclusion of alternative causes of chest pain such as pericarditis, pulmonary embolism, aortic dissection, and hypertrophic cardiomyopathy.

Stress echocardiography is an established techniqufe, largely used for the assessment of known or suspected CAD with high diagnostic and prognostic value [9].

Regarding nuclear imaging, single photon emission computed tomography (SPECT) is most commonly used for clinical myocardial perfusion imaging, whereas positron emission tomography imaging (PET), is the clinical reference standard for the quantification of myocardial perfusion.

In this review, we provide an overview of multimodality imaging approach in patients with ICC, underlying the new development and the future applications of noninvasive imaging. Furthermore, we are going to emphasize the role of each technique based on clinical needs, in order to have a patient-based approach. 

## 2. Computed Tomography Imaging 

### 2.1. Coronary Computed Tomography Angiography

The role of CCTA for management of suspicious CAD is rapidly increasing following the recommendation of European and American guidelines for the evaluation of patients with stable chest pain [1,10].The main advantage of CCTA is the high negative predictive value for CAD evaluation [1].

Due to the low positive predictive value of CCTA for CAD evaluation, CCTA does not represent the best technique for the evaluation of significant CAD [7,11]. Despite this fact, it is possible to obtain information regarding severity of stenosis and plaque composition from CT acquisition [3,6].

In terms of stenosis, CT can provide a qualitative or quantitative assessment of coronary stenosis assigning a final CAD-RADS score [12]. The CAD-RADS score gives evaluation of CAD burden [13], and it is extremely useful for determining the correct therapeutic management of the patient [12]. In terms of prognosis, CAD-RADS score is extremely helpful for risk stratification [14,15]. In a large cohort of patients coming from the multicentric PROMISE study, Bittner showed that the hazard ratio increase from 2.43 of CAD-RADS 1 to 21.84 of CAD-RADS 4b and 5 [14]. Interestingly, Park et al., focused on the prognostic values of CAD-RADS in patients with low-intermediate risk that underwent CCTA in an emergency department for acute chest pain [15]. The authors, using early or late major cardiac events as outcomes, created the three following models: model 1, clinical risk score; model 2, clinical risk score + CAD-RADS; model 3, clinical risk score + CAD extent [15]. The models where CAD-RADS and CAD-extent were added to clinical risk score showed a better prediction for early and late events with a respectively C-index of 0.90 (model 1) and 0.91 (model 2) for early events and 0.89 (model 1) and 0.90 (model 2) for late events [15]. 

In terms of follow up of patients with ICC, CAD-RADS classification may underestimate the progression of disease [16]. Szilveszter et al., in a cohort of 115 patients, evaluated the progression of CAD disease by analyzing the segment stenosis (SSS), segment involvement score (SIS) and CAD-RADS [16]. Data obtained from the analysis with different models showed that it was possible to observe the progression of disease in 53%, 29.6% and 28.7% based on SSS, SIS and CAD-RADS, respectively. [16] In particular, only 54% of patients showed progression based on CAD-RADS score [16]. Despite the importance of CAD-RADS in terms of stenosis classification, treatment planning and prognostic stratification, the data showed that it is necessary to evaluate its role in terms of follow up of patients with ICC.

Plaque analysis represents another important advantage of CCTA over other noninvasive imaging techniques [17].The association between high-risk plaque (HRP) and increased incidence of development of acute coronary syndrome is well known (ACS) [18]. 

CCTA acquisition can provide information about plaque composition. In particular, CCTA can show the presence and extent of positive remodeling index, napkin ring signs, plaque burden, spotty calcification and volume of noncalcified plaque, which are associated with a worse prognosis [18,19]. 

Though analysis of plaque could represent an important role in terms of risk stratification, it is extremely important to consider that quantification of plaque components can be different based on the CT scanner [17] and tube voltage [20]. 

Despite the fact that the quantification of plaque it is not perfectly reliable, it is important to consider the potential application in the future in terms of follow up of patients with ICC. Different studies coming from the PARADIGM study evaluated changes of coronary artery disease [21,22]. In particular, it seems that statins can increase the calcification of the plaques over the time [22]. 

The role of CCTA can be extremely important in the evaluation of patients that underwent percutaneous coronary intervention [23] or coronary artery bypass graft (CABG) [24].

Follow up of intrastent restenosis (ISR) can be extremely difficult but still feasible with CCTA [23]. In particular, due to beam hardening artifacts, it is not uncommon to report false positive results and to send patients to invasive coronary angiography (ICA) considering that beam hardening artifacts is considered significant ISR [23]. However. the new generation CT scanners are improving the possibility of evaluating significant ISR [24,25] increasing the diagnostic accuracy due to higher temporal and spatial resolution [25]. A representative case is shown in Figure 1.

Furthermore, dual-energy technology, using virtual monochromatic images, allows the reduction of streak artifacts and provide iodine maps useful for a better evaluation of intrastent restenosis [26,27,28]. 

In those patients that underwent to CABG, CCTA can be extremely helpful for the evaluation of bypass patency [24,29]. It is possible to evaluate the CABG with CCTA with both high sensitivity and specificity [29].

CCTA, as mentioned above, can be extremely helpful in patients with ICC for the evaluation of coronary anatomy; however, its role is very limited for the evaluation of hemodynamically significant CAD and it could require further examinations such as stress echocardiography (SE), stress CMR, stress CT perfusion (CTP) or noninvasive imaging with nuclear medicine. 

### 2.2. FFRct and CTP

Considering the low positive predictive value of CCTA in the presence of a moderate or severe stenosis, additional noninvasive testing could prove to be useful for the assessment of myocardial ischemia [8]. 

Fractional flow reserve is defined as ratio between the blood pressure distal to a stenosis and the pressure before the stenosis during hyperemia [30]. The application of computational fluid dynamics allows the possibility to calculate FFRct from CCTA images [31]. 

There are several parameters that identify the FFRct as positive after the application of the FFRct algorithm. 

Nowadays, the addition of FFRct on top of CCTA allows for the reclassification of patients that underwent CCTA, thus increasing the positive predictive value for depiction of significant CAD [11,32]. 

This algorithm allows for the reduction of invasive coronary angiography in order to reclassify the patients based on the functional assessment of the plaque over the anatomical coronary stenosis [33]. Beyond the low rate of ICA, the application of FFRct in patients with ICC is extremely helpful considering the prognostic impact in the short term and the low number of major adverse cardiac events (MACE) in patients with FFRct > 0.80 on follow up of one years obtained from literature [34,35]. 

Similar to FFRct, stress CTP, regardless of the static or dynamic acquisition, can be extremely useful for the evaluation of myocardial ischemia over the plaque stenosis [8,12,33]. 

Contrarily to FFRct, the stress CTP requires the administration of a stress agent followed by another CT acquisition in single energy or dual energy [8]. 

The interpretation of static stress CTP is more qualitative or semiquantitative, and it does not allow the evaluation of myocardial blood flow. [8] The latter is evaluated with dynamic stress CTP that allows a quantitative approach [8]. The acquisition of both stress CTP allows the reclassification of patients over the plaque stenosis and allows for the evaluation of myocardial ischemia [11,32]. In particular, it has been demonstrated the increase of diagnostic accuracy over CCTA when stress CTP is applied [11,32]. Additionally, stress CTP can be extremely helpful in patients showing previous history of PCI, thus increasing the diagnostic accuracy over CCTA in both territory and patient-based analysis with results of 97% and 96%, respectively. 

Few studies evaluated the prognostic impact of stress CTP in clinical practice [36,37,38]. 

Meinel et al. demonstrated that segments deficient of stress CTP are strongly associated with a worse prognosis [37] while Van Assen et al. demonstrated that stress CTP has a higher prognostic value compared to CCTA or FFRct for prediction of MACE [36]. 

In the setting of ICC, FFRct could have some limitation in clinical practice in patients with previous PCI and stent implantation [39], but FFRct can play a key role for the evaluation of CAD stenosis in order to increase the diagnostic accuracy for the detection of hemodynamically significant CAD and provide information regarding the prognosis in the short term [34]. While CTP can evaluate the myocardial ischemia in all ICC settings, it is important to consider that compared to FFRct, CTP has a high radiation exposure and it needs the administration of a stress agent [8]. 

A correct approach using FFRct and stress CTP on top of CCTA can definitely provide identification of ischemic coronary plaques and steer the therapeutical approach. 

A representative case of FFRct and myocardial perfusion are shown in Figure 2 and Figure 3, respectively.

## 3. Cardiac Magnetic Resonance

### 3.1. Stress Cardiac Magnetic Resonance

The goal of noninvasive functional tests for the diagnosis of obstructive CAD is to detect myocardial ischemia [1]. CMR can be used to assess ischemia in two ways: either by monitoring changes in left ventricular kinetics after dobutamine administration, or by monitoring myocardial perfusion under stress with vasodilator drugs (e.g., dipyridamole, adenosine or regadenoson). This last technique is the most commonly used and is intended to identify a myocardial perfusion defect, which can be assessed qualitatively or quantitatively [40].

A typical stress-CMR perfusion protocol is characterized by the administration of a vasodilator during contrast media injection with a series of perfusion sequences acquired at the basal, mid-cavity, and apical slices from the short-axis views.

A perfusion defect is defined as a region of subendocardial hypointensity in a coronary distribution. This defect should be evaluated while the myocardium is perfused, rather than immediately upon contrast arrival within the left ventricular cavity (i.e., two to three heartbeats after the peak of contrast media enhancement in the left ventricular cavity) and should last for a few seconds. To differentiate perfusion defects caused by myocardial ischemia or myocardial fibrosis, this assessment must be accompanied by the evaluation of late gadolinium enhancement (LGE) images: the absence of LGE associated with a stress perfusion defect defines the presence of inducible ischemia. The number of segments with perfusion defects may be used to calculate the total ischemic load: according to ESC guidelines [1], the presence of more than two segments with stress perfusion defects indicates a patient at high risk of an event. Along with the visual assessment of the area of inducible ischemia, semiquantitative or fully quantitative techniques have emerged. These analyses are based on an examination of the signal intensity time curves. Semiquantitative assessments are based on a comparison of data collected under stress and data obtained at rest, whereas absolute quantification of myocardial blood flow is based on a mathematical deconvolution of these curves. These techniques are promising, and based on a recent meta-analysis, they achieve sensitivity, specificity and area under the curve of 0.77, 0.84, and 0.87, respectively, for the semiquantitative analysis and 0.77, 0.86 and 0.88, respectively, for the quantitative analysis. Overall, semiquantitative, quantitative stress-CMR perfusion and subsegmental qualitative analyses are a new application for assessing ischemic burden and coronary microvascular function; however, standardization and validation of those techniques are still ongoing and are required before they can be safely used in clinical practice [40,41,42].

First, Schwitter et al. reported in 2008 in the MR-IMPACT, [43] a multicenter and multivendor scenario, that stress-CMR perfusion is a valuable technique for CAD detection and that it could be considered as an alternative to SPECT imaging for the evaluation of selected patients with known or suspected CAD. The CE-MARC trial [44] demonstrated the superiority of stress-CMR perfusion over SPECT a few years later, and the excellent diagnostic performance of the stress-CMR perfusion was also confirmed by the MR-IMPACT II [45] and GadaCad trial [46]. A recent meta-analysis by Pontone et al. [7] reported a sensitivity and specificity of 0.87 and 0.88, respectively, in detecting obstructive CAD at a patient level with a diagnostic odds ratio of 49.36. Based on these findings, we can confidently state that a comprehensive CMR protocol that includes stress perfusion allows for high-accuracy assessment of contractile function, viability, and myocardial ischemia in a single examination.

Moreover, this technique has significant prognostic implications: as reported in the SPINS registry [47], a negative test is associated with a very low incidence of cardiac events, a low need for coronary revascularization, and a low cost of subsequent ischemia testing. In 2019, Nagel et al. reported in MR-INFORM [48] that the use of stress-CMR perfusion prior to ICA reduces the number of ICA procedures, is associated with a lower rate of coronary revascularization than an invasive fractional flow reserve (iFFR)-guided revascularization strategy without an increase in MACE at one year. Furthermore, the CE-MARC 2 trial [49] found that a comprehensive CMR examination reduced the risk of unnecessary ICAs within 12 months in patients with suspected angina compared to NICE guideline-guided treatment, with no statistically significant difference between CMR and SPECT techniques and no increase in MACE. Overall, we can assume that stress-CMR perfusion revascularization has the same efficacy as iFFR-guided perfusion revascularization and may be especially useful in clinically stable individuals with a moderate to high pretest likelihood of significant CAD.

The main disadvantages of stress-CMR perfusion are its low availability and the high level of expertise required. On the other hand, because of its clinical advantages, lack of ionizing radiation, and low cost, it is a radiological technique that will be used more frequently in the future to study myocardial ischemia, especially considering a more quantitative approach. Furthermore, it could represent a useful tool for the functional assessment on top of CCTA in presence of CAD that requires the evaluation of myocardial ischemia. A case of stress CMR is shown in Figure 4.

### 3.2. CMR Tissue Characterization 

Despite tissue characterization with CMR in patients with ICC could be obtained without administration of contrast agent using native T1 mapping [50,51,52], in clinical practice it is mainly confined to the evaluation of LGE for the assessment of myocardial viability [2]. 

LGE sequences are acquired 10–15 min after the administration of gadolinium-based contrast agent [2,53]. In LGE sequences, the normal myocardium is “black” while the damaged myocardium will appear bright [54]. These differences in terms of signal intensity are due to a prolonged washout of gadolinium in the damaged myocardium [54]. 

The standard sequences used for the evaluation of LGE are usually T1-weighted gradient echo acquisition reconstructed in magnitude or phase-sensitive [2,55]. 

Typical pattern of LGE in ischemic cardiomyopathy is represented by subendocardial LGE that can be extent in transmural scar [56]. 

However, sometimes it is difficult to differentiate between scar and myocardial blood considering that both of them can appear hyperintense showing the same signal intensity [2]. In order to overcome this limitation “black blood” LGE (BBLGE) sequences were developed where myocardium, blood pool and scar were respectively grey, black and hyperintense [57]. Using these sequences it is possible have a better evaluation of subendocardial infarction, papillary muscles involvement [57,58]; however, it is important to consider that, using BBLGE it is possible to have an underestimation of transmurality comparing them to standard LGE sequences [58].

In patients with hibernated myocardium, the absence of LGE or presence of subendocardial LGE with <50% of transmurality is considered viable [59]. The “cut off” of 50% of transmurality is considered a well validated predictor of LV functional recovery after PCI and CABG [60,61,62].

In the setting of ICC, it is mandatory to evaluate the presence of LGE considering the important role in terms of LV recovery. 

Despite stress CMR can provide information regarding the myocardial ischemia, the presence of LGE, in the setting of ICC, allows the assessment of infarcted myocardium. This latter information provides the real advantage of CMR over the other techniques such as stress CTP or SE; in particular considering the transmurality of LGE. 

A representative case has shown in Figure 5.

## 4. Echocardiography

### 4.1. Assessment of LV Function at Rest

The most widely accepted echocardiographic parameter for LV systolic function assessment is left ventricle ejection fraction (LVEF), a measure derived from volumes assessment, defined as the ratio of stroke volume (end-diastolic volume–end-systolic volume) and end-diastolic volume. Clinical decision-making and patient management in several cardiovascular conditions primarily rely on LVEF [63,64,65]. Indeed, this parameter represents an important predictor of outcome in heart failure following myocardial infarction and represents an important predictor of sudden cardiac death [66]. LVEF ≤35% is the currently recommended treatment cut-off for ICD implantation for the primary prevention of sudden cardiac death [63]. 

However, technical limitations and the dependence of image quality from the acoustic window, with consequent difficulty in distinguishing between the endocardium and the blood pool, make poor LVEF calculated by echocardiography poorly reproducible. 

In addition, as a volume-derived index, LVEF relies on geometric assumptions (particularly two-dimensional echocardiography), is dependent on load and heart rate, is influenced by changes in LV geometry, and does not correspond to true LV contractility [64]. 

Interestingly, volumes assessed with echocardiography in patients with ICC have an important diagnostic and prognostic value, independent from LVEF. Left ventricular dilatation at echocardiography is associated with a poor prognosis. In particular, a high end-systolic volume index is predictive of heart failure hospitalization inpatient with ICC [67]. 

In clinical routine, LV volume measurements are usually obtained in transthoracic echocardiography using the biplane method of disk summation (modified Simpson’s rule) acquiring left ventricle (LV) volume from apical four- and two-chamber views, taking care to maximize LV area, avoiding apex foreshortening. On the contrary, the image acquisition in three-dimensional (3D) echocardiography should focus primarily on including the entire LV within the pyramidal data set as 3D echocardiography does not rely on geometrical assumption. For this reason, 3D echocardiography is recommended nowadays for volumes measurements when feasible, depending on imaging quality. [68]. Compared with traditional 2D methods, 3D echocardiography is more accurate and comparable to CMR in patients with a good image quality, although volumes tend to be smaller [69]. 

### 4.2. Role of Strain in Ischemic Cardiomyopathy

In order to partially overcome the limits of LVEF, new parameters have been proposed for the assessment of LV function in echocardiography. Between them, the most used are indices of myocardial deformation obtained with speckle-tracking echocardiography, such as strain and strain rate, proposed as adjunctive parameter to LVEF [70]. Strain indicates the tissue deformation (expressed in percentage) during the cardiac cycle, assuming positive strain values for elongation and negative strain values for shortening, while strain rate indicates the speed at which the deformation occurs [70].

As the endocardial layer of the LV wall is more susceptible to ischemic injury in patients with cardiac ischemia, global longitudinal strain (GLS) is more affected than other parameters. It is more effective than LVEF in identifying patients with significant CAD [71]. Biering-Sørensen et al. showed that in patients with suspected stable angina pectoris, GLS assessed at rest is an independent predictor of significant CAD and significantly improves the diagnostic performance of exercise test and seems capable of identifying high-risk patients [72]. 

Notably, the assessment of LV GLS with speckle-tracking echocardiography is related to long-term outcomes in patients with ICC. In a study of 1060 patients with ICC, GLS was independently related to all-cause mortality and heart failure [73].

### 4.3. Stress Echocardiography

Thanks to its ubiquitous nature, portability, relatively low cost, and avoidance of ionizing radiation, SE is an attractive choice of test for a wide variety of patients [38] and has received Class 1A and 1B indication for the evaluation of stable chest pain in European and American guidelines respectively [1,10]

SE is able to detect myocardial ischemia by identification of wall motion abnormalities. Moreover, several prognostic parameters, including blood exercise duration, pressure response, ST-T changes, and LV dilatation, can be simultaneously assessed and enhance its diagnostic and prognostic role. [9]. In terms of diagnostic accuracy, SE seems more indicated for the evaluation of patients with three-vessels disease compared to perfusion imaging techniques [74]. Regarding its prognostic accuracy, SE has a very high negative predictive value (98.8%) for both primary and secondary events [40].

However, the test is dependent on the operator experience, and a poor acoustic window can hinder its diagnostic accuracy. 

Contrast echocardiography can improve image quality, reader confidence, and reproducibility for assessing both LV structure and function. Moreover, in patients with CAD and wall kinesis abnormalities, contrast agents can be used when thrombi are suspected but not clearly documented [75]. In patients undergoing stress echocardiography, contrast echocardiography improves the visualization of regional wall motion abnormalities and increases reader confidence in study interpretation [76]. Moreover, although largely underutilized, myocardial perfusion can improve the detection and risk stratification of patients with CAD using SE. Myocardial contrast echocardiography during SE allows for the assessment of both wall motion and perfusion, enhancing the detection of wall motion abnormalities and the identification of perfusion defects [76]. 

Speckle-tracking echocardiography has also been proposed to reduce the subjectivity of image interpretation in SE. A recent study from Ilardi et al. demonstrated that strain evaluation improves visual wall motion assessment to detect inducible ischemia in the myocardial territories supplied by the left anterior descending artery in patients undergoing dobutamine stress echocardiography [77]. Finally, artificial intelligence applications have been recently applied to rest and stress echocardiography to improve the high inter-observer variability experienced in human evaluation. However, further developments and test cohorts are required for the widespread clinical implementation of AI in real-life echocardiographic workflow for CAD diagnosis [78].

The major advantages of echocardiography are still represented by its low cost, wide availability and lack of ionizing radiation. Therefore, the role of echocardiography, in patients with ICC, is still fundamental in clinical practice allowing the evaluation of biventricular function and evaluation of ischemia. 

A representative case is shown in Figure 6.

## 5. Nuclear Medicine 

### 5.1. Myocardial SPECT 

Myocardial perfusion imaging (MPI) with SPECT is one of the most performed noninvasive cardiac imaging tests. The biggest advantage of SPECT is that there is extensive clinical experience as well as an abundance of studies demonstrating the ability of SPECT in the diagnosis, risk stratification, treatment, and prognostic evaluation for patients with coronary artery disease [79,80]. This technique utilizes a radiotracer (Tc99m-tracers and Tl201) to evaluate the distribution of blood flow and cell membrane or mitochondrial integrity in myocardium during stress and at rest [81]. A meta-analysis of large studies reported an average sensitivity of 87% and a specificity of 73% of SPECT MPI in detecting obstructive CAD demonstrated on coronary angiography [82]. As a functional modality for assessing the hemodynamic significance of coronary lesion, SPECT MPI has also established its role in risk stratification and prognostic assessment in patients with CAD [83,84]. A normal scan has an excellent negative predictive value for future adverse cardiac events, even in higher pretest risk populations, such as those with diabetes and chronic kidney disease [85,86]. Conversely, several studies have shown a clear relationship between the extent and severity of inducible ischemia and prognosis [87,88,89]. An extensive review from a total of 69,655 patients reported the median annual rate of cardiac death or nonfatal myocardial infarction was 5.9% for those with high-risk SPECT results. For patients with normal perfusion results, the median rate of cardiac death or nonfatal myocardial infarction was 0.6% per year [90]. Despite the diagnostic performance, underestimating the true extent of obstructive epicardial coronary disease (e.g., situations with balanced ischemia) and the inability to determine microvascular dysfunction [91] remain major limitations of stress with conventional SPECT MPI. Another limitation is the study in high-BMI patients, but using new detectors for SPECT allows the quantification of myocardial blood flow and is now also suited to patients with a high BMI [92].

Several perfusion and nonperfusion markers have been proposed to improve detection of left main or three-vessel coronary disease [84,93]. Risk assessment with SPECT MPI is improved if functional assessments from ECG-gated SPECT measurements of ejection fraction and end systolic volume are considered. Additionally, transient ischemic dilation of the left ventricle can be evaluated. [94] Each of these measurements are important nonperfusion predictors of severe CAD [95,96]. The demonstration by SPECT imaging of significant, inducible myocardial ischemia to indicate viability, not only defines a high cardiovascular risk for a specific patient, but can also guide treatment to improve outcome. Observational studies with long-term follow-up suggested that ischemia on SPECT-MPI could identify high-risk patients who might have reduced mortality with early revascularization compared to medical therapy [97,98,99]. Recently, there is a trend toward shorter acquisition times and reduced radiation exposure for SPECT MPI [100]. Alternative imaging modalities and improved reconstruction algorithms have emerged for this purpose and a new solid-state detector technology [101], such as cadmium-zinc-telluride detectors, has been introduced for dedicated cardiac SPECT systems (CZT SPECT) [102]. Moreover, CZT SPECT through fast dynamic tomography provides some new parameters for quantitative analysis of absolute myocardial blood flow (MBF) and myocardial flow reserve (MFR), which may improve the detective sensitivity and avoids missed diagnosis or underestimation [103,104]. A representative case has shown in Figure 7.

### 5.2. Myocardial PET

While less available and more expensive than SPECT, PET offers higher sensitivity and spatial resolution [105]. The most commonly used PET perfusion tracers used for conventional PET MPI are ^82^Rb-chloride and^13^N-ammonia, with a small number of centers worldwide using ^15^O-water [106]. A meta-analysis to compare the diagnostic accuracy of cardiac PET and SPECT for the evaluation of patients with known or suspected CAD using coronary angiography as the reference standard demonstrated higher pooled sensitivity with PET (92.6%) compared with SPECT (88.3%) [107]. PET MPI has expanded significantly over the past decade. With its ability to assess contractile function abnormalities with gated imaging and quantify global and regional myocardial blood flow (MBF) [108], PET MPI is a powerful tool to assess patients with coronary artery disease (CAD) [109]. Measurement of MBF is validated for most PET perfusion tracers, although the ideal tracer for MBF quantitation is [^15^O]-water [106]. Subsequent studies including quantitative perfusion data showed an excellent diagnostic performance of the measurements for CAD detection and indicated that MBF and myocardial flow reserve (MFR) have an added value over the visual assessment of perfusion [110,111], including that it allows for detection of microvascular disease and three-vessel disease, or left main disease [112,113]. The European Society of Cardiology (ESC) has recently published new guidelines on the diagnosis and management of chronic coronary syndromes (CCS) where a new concept of “clinical likelihood of CAD” was introduced accounting for the impact of various risk factors and modifiers on the pretest probability [1]. When available, imaging tests are recommended as the initial strategy to diagnose CAD in symptomatic patients. Knuuti et al. demonstrated good performance of PET imaging and stress CMR with optimal application ranges for both ruling-in and ruling-out disease for anatomic and functional CAD [114]. Noninvasive assessment of global coronary flow reserve (CFR) has emerged as a noninvasive, quantitative prognostic marker of cardiovascular risk [115,116,117]. In patients with suspected or overt ischemia, abnormal perfusion PET was associated with a higher incidence of MACE and cardiac death. In patients with normal perfusion imaging, abnormal CFR was independently associated with a higher annual event rate over three years compared with normal CFR. In patients with abnormal perfusion imaging, an impaired CFR has added value for predicting adverse outcomes [118,119]. Similarly, the decrease in MFR was found to be a more sensitive predictor for cardiac death than LVEF in patients with chronic CAD submitted to PET-driven revascularization [120]. Noninvasive PET modality with (^18^F)-fluorine-deoxyglucose (FDG) has become a standard for myocardial viability assessment [121,122]. A meta-analysis reported superior accuracy of PET and CMR imaging for assessment of myocardial viability compared to SPECT MPI and dobutamine echo; the weighted sensitivity and specificity of (F18-)FDG PET to predict functional recovery after revascularization was of 92% and 63%, with a PPV and NPV of 74% and 87%, respectively [123]. The potential of this imaging technique has been further increased with the introduction of hybrid scanners (PECT/CT and PET/MR) [124,125] as well as new radionuclides offering new applications for improving management of patients with chronic CAD [126]. In order to have visualization of perfusion defect in a similar way to SPECT, it is possible to consider different tracers such as the ^13^N-ammonia. Absolute MBF measurements can be provided using tracer kinetic modelling and correction for limited extraction. However, one of the limits of this tracer is its short half-life (9 min) that requires the presence of an on-site cyclotron.

Stress imaging with nuclear medicine is widely 
available in clinical practice, allowing the evaluation of myocardial ischemia. 
Although stress CTP, myocardial SPECT and myocardial PET require the use of 
ionizing radiation, it is important to consider that with CTP it is possible to 
evaluate CAD beyond myocardial ischemia. Compared to SE and stress CMR, using 
nuclear medicine it is possible to evaluate ischemic myocardium, however it is 
mandatory to consider the use of radiation. A representative case is shown in Figure 8.

## 6. Future Perspective 

Nowadays a multimodality approach represents a fundamental tool for an appropriate management of patients with ICC. The multimodality approach in patients with ICC allows to develop different strategies based on the clinical history of patients and local expertise (Figure 9). 

The possibility to have a comprehensive and detailed evaluation of coronary plaque anatomy together with assessment of myocardial ischemia allows the possibility to have the best therapeutical approach. Indeed, the sequential strategy, due to a multimodality approach, can be extremely helpful for the correct identification, diagnostic pathway, providing a customized approach based on clinical needs. 

The future of integrated approach (anatomical and functional) in ICC will be revolutionized by the revolution of Artificial Intelligence applied to medical imaging [127]. Several studies were published highlighting the possibility to obtain evaluation of coronary artery stenosis, plaque composition, LGE, volumes and nuclear medicine perfusion in short time using AI algorithm [128,129,130,131,132,133]. Using AI in the future will make it possible to obtain information regarding the status of patients with ICC in a relative short time with high precision; furthermore, it would be possible to obtain a prognostic stratification of these patients [134,135,136].

Radiomics is another interesting topic that will be further developed in the field of cardiovascular imaging [137,138]. Using radiomics, it is possible to evaluate some components of images that could be extremely helpful for a better evaluation of myocardial and coronary plaque tissue. 

Considering the aforementioned techniques, noninvasive imaging in patients with ICC is moving to a precision-based approach with a customized treatment, struggling to avoid unnecessary invasive procedures. Another interesting topic that should be underlined in patients with ICC, is the mandatory use of imaging for planning of the best surgical approach and selection of coronaries that should be revascularized. Therefore, it should not surprise that patients will undergo to several imaging techniques before CABG [139].

An important issue that should be further explored in clinical practice is represented by the prognostic significance of plaque analysis versus ischemia for development acute cardiac events. Following the results of ISCHEMIA trial [140], it could be important to consider carefully the best therapeutic approach without exclude in the future a pivotal role of plaque imaging for identification of patients that could develop acute events.

Despite the role of noninvasive imaging in patients with ICC can be complementary, it is not possible to ignore that in prognostic stratification some information obtained with noninvasive techniques can have different importance.

In the future, in spite of multimodality imaging’s role in ICC, it will be extremely important to consider the balance between local expertise, clinical needs and economical sustainability. Considering the new ESC [1] and ACC [10] guidelines an adequate training of healthcare personnel involved in noninvasive cardiac imaging will be fundamental for a correct management of patients with ICC [141]. The clinical need is for safer examination. Therefore, in clinical practice it could be useful to exclude some techniques to avoid health risks for the patients. Finally, it will be important to consider the economic impact of each technique in the community. 

In conclusion, a multimodality approach of patients with ICC is fundamental in the modern era for a better evaluation and follow up of CAD. 

In the future, the combination of data obtained with noninvasive multimodality imaging with AI, radiomics, clinical information according to sustainability and local expertise could play a key role for a better and faster classification and management of these patients. 

## Figures and Tables

**Figure 1 jimaging-08-00035-f001:**
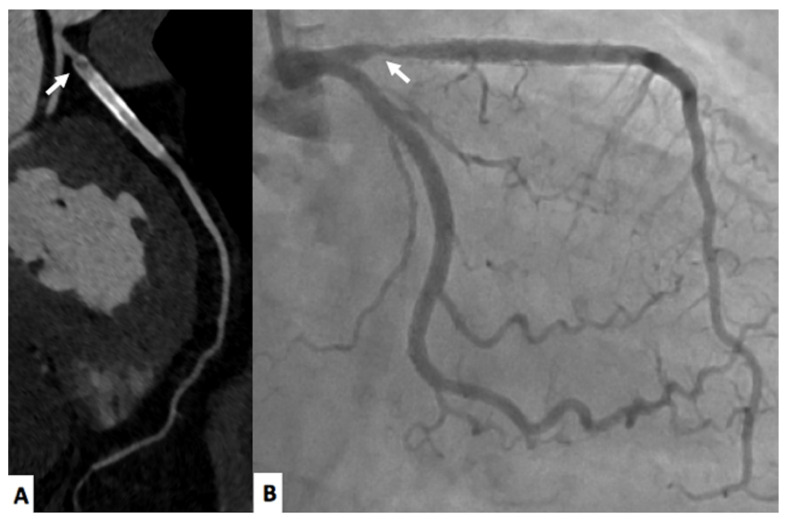
A 77-year-old male with in-stent restenosis in the proximal segment of the left anterior descending artery (**A**), a filling defect in the proximal part of the stent (arrow) is clearly visualized. The invasive coronary angiogram (**B**) shows in-stent restenosis in the stent located in the proximal left anterior descending artery (arrow).

**Figure 2 jimaging-08-00035-f002:**
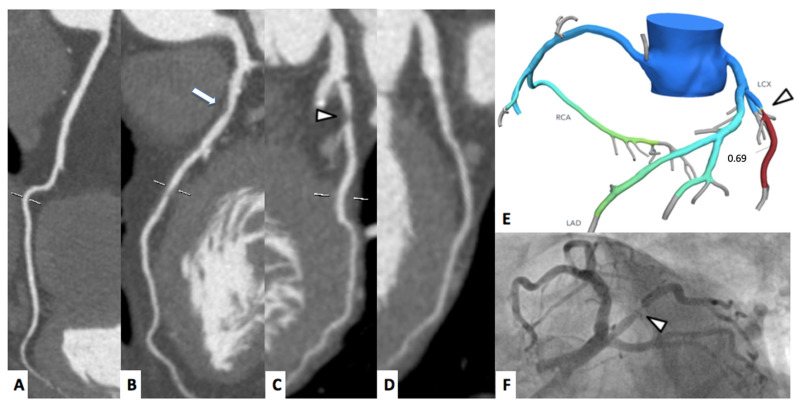
A 70-year-old female patient underwent to CCTA for chest pain. Right coronary artery (**A**) and left circumflex artery (**D**) were free from significant stenosis; whereas left anterior descending artery shows moderate proximal stenosis ((**B**), arrow). Ramus intermedius shows a severe, fibro-lipid plaque stenosis ((**C**), arrowhead). The FFR_CT_ assessment confirmed the functional significance of the stenosis proximal-ramus intermedius ((**E**), arrowhead), whereas the FFR_CT_ values of the left anterior descending artery were above the ischemia threshold of 0.80. The invasive coronary angiogram ((**F**), arrowhead) shows severe stenosis of the proximal tract of the ramus intermedius.

**Figure 3 jimaging-08-00035-f003:**
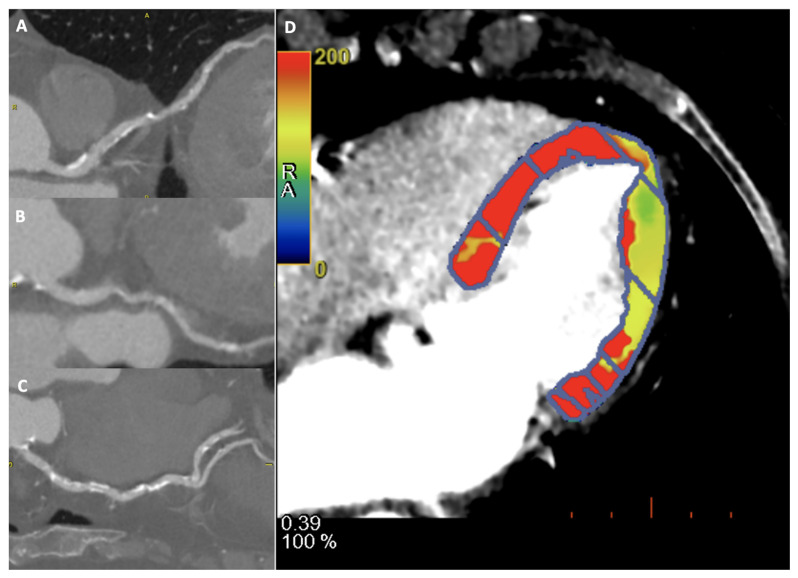
A 60-year-old female patient underwent to CCTA and dynamic CT perfusion for suspected coronary artery disease. Diffuse coronary calcification on left anterior descending artery (**A**), left circumflex (**B**) and right coronary (**C**), showing mild reduction of flow in dynamic CT perfusion on lateral wall, in particular at the apical segment (**D**).

**Figure 4 jimaging-08-00035-f004:**
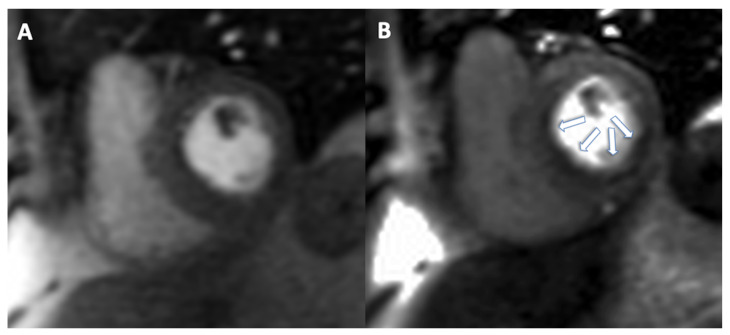
A 65-year-old male patient with known history of chronic total occlusion of right coronary artery, moderate stenosis on left anterior descending artery and mild stenosis on right coronary artery. Rest perfusion sequence does not show any significant defect of perfusion (**A**). During stress acquisition, a deficit of perfusion was observed in the septum and inferior wall ((**B**), arrows).

**Figure 5 jimaging-08-00035-f005:**
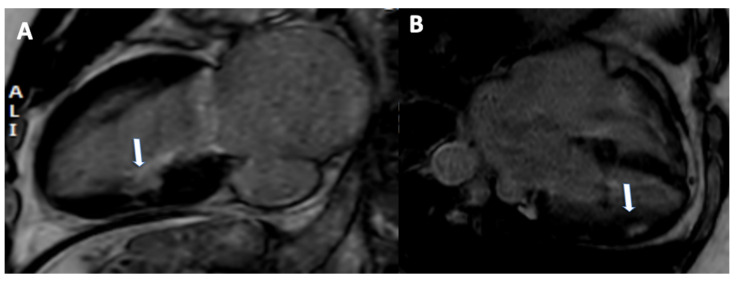
A 45-year-old patient with history of myocardial infarction. Subendocardial late gadolinium enhancement was observed in inferior (arrow, (**A**)) and inferolateral (arrow, (**B**)).

**Figure 6 jimaging-08-00035-f006:**
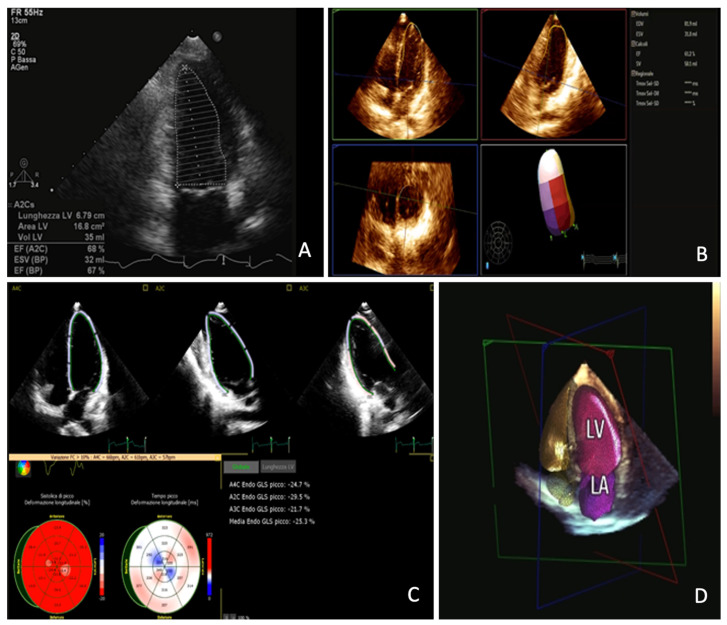
Panel (**A**): bi-dimensional echocardiography measurement of left ventricle ejection fraction; Panel (**B**): three-dimensional echocardiography measurement of left ventricle ejection fraction; Panel (**C**): Left Ventricle global longitudinal strain assessment; Panel (**D**): three-dimensional left ventricle and left atrium automatic measurement with artificial intelligence algorithm.

**Figure 7 jimaging-08-00035-f007:**
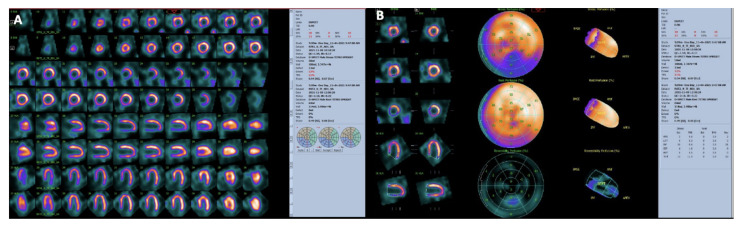
Myocardial perfusion SPECT with Tc-99m-Tetrofosmin in a patient with CAD performed at stress (maximal cycloergometer exercise) and at rest. Stress and rest slices in three axes are shown in (Panel (**A**)). Polar maps under stress and at rest are shown in (Panel (**B**)). Slices and polar maps show a stress perfusion defect in the inferior wall which significantly improves (reverses) in the rest study. The pattern is typical of stress-induced myocardial ischemia.

**Figure 8 jimaging-08-00035-f008:**
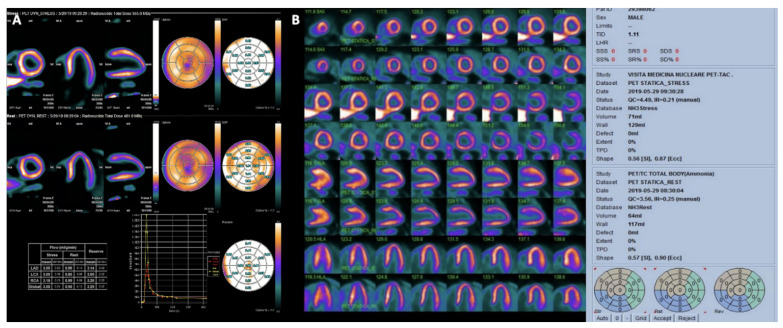
(Panel (**A**)) dynamic myocardial 13N-ammonia PET/CT in a normal patient. First row shows representative left ventricular perfusion myocardial slices under maximal vasodilator stress (with quantitative myocardial stress flow values); second row shows representative left ventricular perfusion myocardial slices at rest (with quantitative myocardial rest flow values). Absolute flow values are presented in the table and flow reserve is automatically calculated for the different territories and also for the whole left ventricle (global). (Panel (**B**)) myocardial 13N-ammonia PET/CT stress/rest perfusion study in the same patient. The distribution of perfusion is homogeneous both at stress and at rest.

**Figure 9 jimaging-08-00035-f009:**
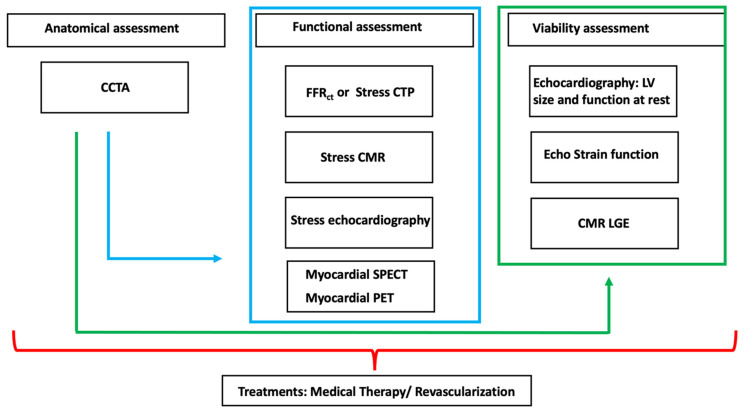
Multimodality evaluation of patients with ICC. Green arrows shows the sequential evaluation of coronary anatomy followed by the assessment of myocardial viability. Blue arrow shows sequential approach for the anatomical evaluation followed by functional assessment. Each approach should be customized considering the local expertise and the clinical needs.

## Data Availability

Not applicable.

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
