# Peer review of "Multimodality Imaging in Ischemic Chronic Cardiomyopathy"

_2313-433X, 2022, doi:10.3390/jimaging8020035_

Round 1

Reviewer 1 Report

General comments:

The authors present a complete summary of available diagnostic techniques in ischemic chronic cardiomyopathy.  The paper is well written and structured, but some adjustments are needed. This reviewer suggests a minor revision about the manuscript.

Specific comments:

TITLE, ABSTRACT, KEYWORDS: Ok.

COMPUTED TOMOGRAPHY IMAGING:

  • Lines 104-106: correct the text as follow: “model 1, clinical risk factors; model 2, clinical risk factors and CAD-RADS scores; model 3, clinical risk factors and extent of CAD”;
  • Line 127: “Thoughanalysis” separate words;
  • Lines 138-143: could be useful to introduce the improvements of Dual-Energy CT (DECT) Virtual Monochromatic Images (VMI) reconstructions in solving problems such as calcium blooming, beam hardening, metal or streak artefacts and in evaluation of in-stent restenosis. (PMID: 31786506; PMID: 30919651) and emerging role of Iodine Maps (PMID: 31925704);
  • Line 140: insert acronym description of ICA;
  • Line 169: please add acronym explanation of MACE at its first use, and remove it at line 260.

FIGURE 2

Please order images or text in the caption to be more easily understood by the reader and highlight stenotic tracts.

CARDIAC MAGNETIC RESONANCE

  • Line 226: add acronym explanation of LGE at first use;
  • Lines 227-230: many studies evidence that more numerous and smaller segments revealed greater accuracy in the evaluation of ischemic myocardium, even if not standardized this information could be useful to the reader. (PMID: 32028980; PMID: 32612708);
  • Line 281: there is evidence that acquisition time and contrast dose can be optimized, studies about protocols optimization should be cited (PMID: 29110679);
  • The role of T1 mapping in tissue characterization could be mentioned (e.g., PMID: 29447739; PMID: 27899132).

ECHOCARDIOGRAPHY

  • Line 349: “Biering-Sørensen” please remove reference’s link;
  • Lines 364-367: there is no correspondence between text and reference 67, please revise;
  • Line 390: correct “has shown” with “is shown”.

NUCLEAR MEDICINE

Some references are too old, please include more recent articles (e.g., PMID: 32284108).

CONCLUSION: Fine.

Author Response

General comments:

The authors present a complete summary of available diagnostic techniques in ischemic chronic cardiomyopathy.  The paper is well written and structured, but some adjustments are needed. This reviewer suggests a minor revision about the manuscript.

 Specific comments:

 TITLE, ABSTRACT, KEYWORDS: Ok.

 COMPUTED TOMOGRAPHY IMAGING:

Q1_R1: Lines 104-106: correct the text as follow: “model 1, clinical risk factors; model 2, clinical risk factors and CAD-RADS scores; model 3, clinical risk factors and extent of CAD”;

A1_R1: We thank the reviewer for the comment. We revised the manuscript accordingly.

Q2_R1: Line 127: “Thoughanalysis” separate words;

A2_R1: We thank the reviewer for the comment. The manuscript was revised accordingly.

Q3_R1: Lines 138-143: could be useful to introduce the improvements of Dual-Energy CT (DECT) Virtual Monochromatic Images (VMI) reconstructions in solving problems such as calcium blooming, beam hardening, metal or streak artefacts and in evaluation of in-stent restenosis. (PMID: 31786506; PMID: 30919651) and emerging role of Iodine Maps (PMID: 31925704);

  • Line 140: insert acronym description of ICA;
  • Line 169: please add acronym explanation of MACE at its first use, and remove it at line 260.

A3_R1: We thank the reviewer for the comment; we agree that few sentences regarding the application of dual energy technology are mandatory in the manuscript. Following the suggestions of the reviewer we add the following paragraph to the manuscript: Furthermore dual energy technology, using virtual monochromatic images, allows the reduction of streak artifacts and provide iodine maps useful for a better evaluation of intrastent restenosis [27-29].

Following the suggestion of the reviewer we descript the words ICA and MACE in our first description. 

FIGURE 2

Q4_R1: Please order images or text in the caption to be more easily understood by the reader and highlight stenotic tracts.

 A4_R1: We thank the reviewer for the comment. We agree that figure 2 need improvements in terms of figure caption and imags. Following the suggestion of the reviewer we modified the figure and the caption as follow: 70 year-old female patient underwent to CCTA for chest pain.  Right coronary artery (A) and left circumflex artery (D) were free from significant stenosis; whereas left anterior descending artery shows moderate proximal stenosis (B, arrow). Ramus intermedius shows a severe, fibro-lipid plaque stenosis (C, arrowhead). The FFRCT assessment confirmed the functional significance of the stenosis proximal - ramus intermedius (E, arrowhead), whereas the FFRCT values of the left anterior descending artery were above the ischemia threshold of 0.80. The invasive coronary angiogram (F, arrowhead) shows severe stenosis of the proximal tract of the ramus intermedius.

CARDIAC MAGNETIC RESONANCE

Q5_R1: Line 226: add acronym explanation of LGE at first use;

A5_R1: We thank the reviewer for the comment. The explanation of LGE has been inserted in the manuscript.

Q6_R1: Lines 227-230: many studies evidence that more numerous and smaller segments revealed greater accuracy in the evaluation of ischemic myocardium, even if not standardized this information could be useful to the reader. (PMID: 32028980; PMID: 32612708);

  • Line 281: there is evidence that acquisition time and contrast dose can be optimized, studies about protocols optimization should be cited (PMID: 29110679);
  • The role of T1 mapping in tissue characterization could be mentioned (e.g., PMID: 29447739; PMID: 27899132).

A6_R1: We agree with reviewer that few words should be mentioned in the manuscript regarding the possibility to evalutate the perfusion using subsegmental analysis and perform dedicated acquisition for both and LGE. Furthermore we agree that T1 mapping could have a limited role in patients with ICC. Therefore we decided to modify the manuscript following the suggestion of reviewer and insert the reference in the manuscript.

ECHOCARDIOGRAPHY

Q7_R1: Line 349: “Biering-Sørensen” please remove reference’s link;

A7_R1: We thank the reviewer for the comment. We modified the manuscript accordingly.

Q8_R1: Lines 364-367: there is no correspondence between text and reference 67, please revise;

A8_R1: We thank the reviewer for the comment. Following the suggestion we modified the manuscript comparing the images of echocardiography with perfusion imaging. We modified the manuscript as it follows: In terms of diagnostic accuracy, SE seems more indicated for the evaluation of patients with three-vessels disease compared to perfusion imaging techniques [76]

Q9_R1: Line 390: correct “has shown” with “is shown”.

A9_R1:  We thank the reviewer for the comment. The manuscript has been modified accordingly.

NUCLEAR MEDICINE

Q10_R1: Some references are too old, please include more recent articles (e.g., PMID: 32284108).

A10_R1: We thank the reviewer for the comment. Following the suggestion of reviewer we modified the manuscript accordingly and we insert in the manuscript some recent articles.

CONCLUSION: Fine.

Reviewer 2 Report

The title of the article indicates that it is a review of the application of multimodality imaging in the diagnosis and management of Ischemic Chronic Cardiomyopathy.

First, in my understanding, multimodality imaging refers to the simultaneous production of signals for more than one imaging modality to improve the imaging quality/early detection, etc. The authors reviewed the application of different imaging modalities in Ischemic Chronic Cardiomyopathy. However, I don’t see much of the review of hybrid imaging modalities(multimodality) other than in part 5: 5. Nuclear Medicine.

Even if the authors attempt to review the incorporation of two or more imaging modalities in a sequential manner, which is also considered as the concept of multimodality imaging. I still don’t see much of the related content.  

Second, the abstract of the paper is more of listing the facts rather than what an abstract is supposed to do: a summary of the review that provides the readers with the big picture of what content/scope is covered in this article.

Third, the review is overall lacks summarization. The contents are not very well-organized. In addition, besides listing facts and results, should the authors consider including more of their opinions/insights/perspectives on this specific topic?

Lastly, spelling/formatting issues, several examples:

Line 6 “Sandro and MD10,11 6”

Page 9, line 306, 4. ECHOCARDIOGRAPHY. The format and font of this subsection is not consistent with others

Page 9, line 338. Role of strain in ischemic cardiomyopathy. Does it belong to part 4?

I would suggest numbering the sub-sub sections as 2.1 Coronary computed tomography angiography, 2.2 FFRct, and CTP…. for the convenience of the readers.

Author Response

The title of the article indicates that it is a review of the application of multimodality imaging in the diagnosis and management of Ischemic Chronic Cardiomyopathy.

Q1_R2: First, in my understanding, multimodality imaging refers to the simultaneous production of signals for more than one imaging modality to improve the imaging quality/early detection, etc. The authors reviewed the application of different imaging modalities in Ischemic Chronic Cardiomyopathy. However, I don’t see much of the review of hybrid imaging modalities(multimodality) other than in part 5: 5. Nuclear Medicine.

Even if the authors attempt to review the incorporation of two or more imaging modalities in a sequential manner, which is also considered as the concept of multimodality imaging. I still don’t see much of the related content.  

A1_R2: We appreciate the reviewer for the comment. In our manuscript the multimodality approach is defined as a possible sequential strategy of non-invasive imaging for the evaluation of ICC. Based on clinical needs of the patients and local expertise, the clinicians should choose the best approach for each patient.

We agree with reviewer that it would be better to specify more in details this kind of approach in the manuscript. In order to have a more readable content, following the suggestion of the reviewer, we revised the manuscript accordingly and we decided to write in each section of the manuscript our opinion and the reason that justify the role of each technique in the diagnostic pathway.

Q2_ R2: Second, the abstract of the paper is more of listing the facts rather than what an abstract is supposed to do: a summary of the review that provides the readers with the big picture of what content/scope is covered in this article.

A2_R2: We thank the reviewer for the comment. Following the suggestion of the reviewer we modified the manuscript accordingly. In particular we add the paragraph at the end of the abstract specifying the aim of our manuscript. The final abstract was modified as follow: In this review we aim to provide an overview regarding the different non-invasive imaging techniques for the evaluation of ICC, providing information ranging from the anatomical assessment of coronary artery arteries to the assessment of ischemic myocardium and myocardial infarction. In particular this review is going to show the different non-invasive approaches based on the specific clinical history of patients with ICC.    

Q3_R2: Third, the review is overall lacks summarization. The contents are not very well-organized. In addition, besides listing facts and results, should the authors consider including more of their opinions/insights/perspectives on this specific topic?

A3_R2: We appreciate the reviewer for the comment. Following the suggestion of the reviewer at the end of each paragraph we summarize the clinical indication of the techniques and we provide our suggestion. We totally agree that our suggestion must be provided in the manuscript.  

Q4_R2: Lastly, spelling/formatting issues, several examples:

Line 6 “Sandro and MD10,11 6”

Page 9, line 306, 4. ECHOCARDIOGRAPHY. The format and font of this subsection is not consistent with others

Page 9, line 338. Role of strain in ischemic cardiomyopathy. Does it belong to part 4?

I would suggest numbering the sub-sub sections as 2.1 Coronary computed tomography angiography, 2.2 FFRct, and CTP…. for the convenience of the readers.

A4_R2: We thank the reviewer for the comment. Following the suggestions we modified the manuscript accordingly. In order to have a more readable manuscript.

Round 2

Reviewer 2 Report

The revised version reflected the authors’ effort in improving the quality of the manuscript. The authors labeled each sections and subsections in a more readable fashion. Some summarizations of clinical indications and authors’ comment were added to each section.

My most concern is remaining the same. In the previous round of review, I raise the question 1 as follow:

“First, in my understanding, multimodality imaging refers to the simultaneous production of signals for more than one imaging modality to improve the imaging quality/early detection, etc. The authors reviewed the application of different imaging modalities in Ischemic Chronic Cardiomyopathy. However, I don’t see much of the review of hybrid imaging modalities(multimodality) other than in part 5: 5. Nuclear Medicine.

Even if the authors attempt to review the incorporation of two or more imaging modalities in a sequential manner, which is also considered as the concept of multimodality imaging. I still don’t see much of the related content.’’

The authors addressed as follow:

“We appreciate the reviewer for the comment. In our manuscript the multimodality approach is defined as a possible sequential strategy of non-invasive imaging for the evaluation of ICC. Based on clinical needs of the patients and local expertise, the clinicians should choose the best approach for each patient.

We agree with reviewer that it would be better to specify more in details this kind of approach in the manuscript. In order to have a more readable content, following the suggestion of the reviewer, we revised the manuscript accordingly and we decided to write in each section of the manuscript our opinion and the reason that justify the role of each technique in the diagnostic pathway.’’

It is appropriate to review “possible sequential strategy” to evaluate ICC to reflect the current advances in multimodality imaging in ICC. However, in the revised version, I don’t see much of related content regarding “sequential strategy”.  The authors listed different imaging applications in ICC. But in terms of sequential strategy is still lacking.

Should the authors consider summarize the diagnostic algorithm or workflow to provide audience an overall picture in addition to goes into details of each individual imaging modalities to matching the title of “Multimodality Imaging in Ischemic Chronic Cardiomyopathy’’

In addition, the spelling and grammatic issue is still existing. Some are very obvious, please check accordingly

For example: page 1, line 10, “The Nethrlands”

Author Response

Question 1: The revised version reflected the authors’ effort in improving the quality of the manuscript. The authors labeled each sections and subsections in a more readable fashion. Some summarizations of clinical indications and authors’ comment were added to each section.

My most concern is remaining the same. In the previous round of review, I raise the question 1 as follow:

“First, in my understanding, multimodality imaging refers to the simultaneous production of signals for more than one imaging modality to improve the imaging quality/early detection, etc. The authors reviewed the application of different imaging modalities in Ischemic Chronic Cardiomyopathy. However, I don’t see much of the review of hybrid imaging modalities(multimodality) other than in part 5: 5. Nuclear Medicine.

Even if the authors attempt to review the incorporation of two or more imaging modalities in a sequential manner, which is also considered as the concept of multimodality imaging. I still don’t see much of the related content.’’

The authors addressed as follow:

“We appreciate the reviewer for the comment. In our manuscript the multimodality approach is defined as a possible sequential strategy of non-invasive imaging for the evaluation of ICC. Based on clinical needs of the patients and local expertise, the clinicians should choose the best approach for each patient.

We agree with reviewer that it would be better to specify more in details this kind of approach in the manuscript. In order to have a more readable content, following the suggestion of the reviewer, we revised the manuscript accordingly and we decided to write in each section of the manuscript our opinion and the reason that justify the role of each technique in the diagnostic pathway.’’

It is appropriate to review “possible sequential strategy” to evaluate ICC to reflect the current advances in multimodality imaging in ICC. However, in the revised version, I don’t see much of related content regarding “sequential strategy”.  The authors listed different imaging applications in ICC. But in terms of sequential strategy is still lacking.

Should the authors consider summarize the diagnostic algorithm or workflow to provide audience an overall picture in addition to goes into details of each individual imaging modalities to matching the title of “Multimodality Imaging in Ischemic Chronic Cardiomyopathy’’.

In addition, the spelling and grammatic issue is still existing. Some are very obvious, please check accordingly

For example: page 1, line 10, “The Nethrlands”

Answer 1: We thank the reviewer for the comment and we agree that a figure should simplify the reading of the manuscript. Following the suggestion of the reviewer we summarized the multimodality approach in figure 9. Based on the clinical need and local expertise it is possible to choose different modalities.

We thank the reviewer for the suggestion regarding the typos and language. Following the suggestion  we fixed them. Regarding linguistic revision the manuscript has been revised by the author LJM, who is native speaker. It will be extremely helpful if the reviewer can indicate in detail the sentences that should be revised.

Round 3

Reviewer 2 Report

The manuscript is much improved and suitable for publication after revision.

Author Response

We thanks the reviewer for the comment. We really appreciated the comment.